# Learning to Sample with Local and Global Contexts from Experience Replay Buffers

**Youngmin Oh[1], Kimin Lee[2], Jinwoo Shin[3], Eunho Yang[3,4], Sung Ju Hwang[3,4]**

[1] Samsung Advanced Institute of Technology
[2] University of California, Berkeley
[3] Korea Advanced Institute of Science and Technology
[4] AITRICS

```
youngmin0.oh@samsung.com,kiminlee@berkeley.edu,
{jinwoos, eunhoy, sjhwang82}@kaist.ac.kr
```

## Abstract

Experience replay, which enables the agents to remember and reuse experience from the past, has played a significant role in the success of off-policy reinforcement learning (RL). To utilize the experience replay efficiently, the existing sampling methods allow selecting out more meaningful experiences by imposing priorities on them based on certain metrics (e.g. TD-error). However, they may result in sampling highly biased, redundant transitions since they compute the sampling rate for each transition independently, without consideration of its importance in relation to other transitions. In this paper, we aim to address the issue by proposing a new learning-based sampling method that can compute the *relative* importance of transition. To this end, we design a novel permutation-equivariant neural architecture that takes contexts from not only features of each transition (local) but also those of others (global) as inputs. We validate our framework, which we refer to as Neural Experience Replay Sampler (NERS)[1], on multiple benchmark tasks for both continuous and discrete control tasks and show that it can significantly improve the performance of various off-policy RL methods. Further analysis confirms that the improvements of the sample efficiency indeed are due to sampling diverse and meaningful transitions by NERS that considers both local and global contexts.

## 1 Introduction

Experience replay (Mnih et al., 2015), which is a memory that stores the past experiences to reuse them, has become a popular mechanism for reinforcement learning (RL), since it stabilizes training and improves the sample efficiency. The success of various off-policy RL algorithms largely attributes to the use of experience replay (Fujimoto et al., 2018; Haarnoja et al., 2018a;b; Lillicrap et al., 2016; Mnih et al., 2015). However, most off-policy RL algorithms usually adopt a unique random sampling (Fujimoto et al., 2018; Haarnoja et al., 2018a; Mnih et al., 2015), which treats all past experiences equally, so it is questionable whether this simple strategy would always sample the most effective experiences for the agents to learn.

Several sampling policies have been proposed to address this issue. One of the popular directions is to develop rule-based methods, which prioritize the experiences with pre-defined metrics (Isele & Cosgun, 2018; Jaderberg et al., 2016; Novati & Koumoutsakos, 2019; Schaul et al., 2016). Notably, since TD-error based sampling has improved the performance of various off-policy RL algorithms (Hessel et al., 2018; Schaul et al., 2016) by prioritizing more meaningful samples, i.e., high TD-error, it is one of the most frequently used rule-based methods. Here, TD-error measures how unexpected the returns are from the current value estimates (Schaul et al., 2016).

However, such rule-based sampling strategies can lead to sampling highly biased experiences. For instance, Figure 1 shows randomly selected 10 transitions among 64 transitions sampled using certain

---

[1]Code is available at https://github.com/youngmin0oh/NERS

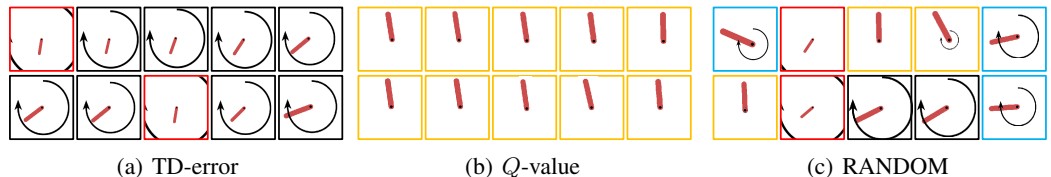

Figure 1: Sampled transitions on Pendulum-v0 from various sampling strategies: (a) Sampling by TD-error, (b) Sampling by $Q$-value, (c) Sampling uniformly at random. Samples highlighted in black, orange, and cyan boxes denote that their state has the rod in downward, upright, and horizontal positions with appropriate amount of actions, respectively. Samples in red boxes have excessively large actions.

metrics/rules under a policy-based learning, soft actor critic (SAC) (Haarnoja et al., 2018a), on Pendulum-v0 after 30,000 timesteps, which goal is to balance the pendulum to make it stay in the upright position. We observe that sampling by the TD-error alone mostly selects initial transitions (see Figure 1(a)), where the rods are in the downward position, since it is difficult to estimate $Q$-value on them. Conversely, the sampled transitions by $Q$-value describe rods in the upright position (see Figure 1(b)), which will provide high returns to agents. Both can largely contribute to the update of the actor and critic since the advantage term and mean-square of TD-errors are large. Yet, due to the bias, the agent trained in such a manner will mostly learn what to do in a specific state, but will not learn about others that should be experienced for proper learning of the agent. Therefore, such biased (and redundant) transitions may not lead to increased sample efficiency, even though each sampled transition may be individually meaningful.

On the other hand, focusing only on the diversity of samples also has an issue. For instance, sampling uniformly at random is able to select out diverse transitions including intermediate states such as those in the red boxes of Figure 1(c), where the rods are in the horizontal positions which are necessary for training the agents as they provide the trajectory between the two types of states. However, the transitions are occasionally irrelevant for training both the policy and the $Q$ networks. Indeed, states in the red boxes of Figure 1(c) possess both low $Q$-values and TD-errors. Their low TD-errors suggest that they are not meaningful for the update of $Q$ networks. Similarly, low $Q$-values cannot be used to train the policy what good actions are.

Motivated by the aforementioned observations, we aim to develop a method to sample both diverse and meaningful transitions. To cache both of them, it is crucial to measure the relative importance among sampled transitions since the diversity should be considered in them, not all in the buffer. To this end, we propose a novel neural sampling policy, which we refer to *Neural Experience Replay Sampler (NERS)*. Our method learns to measure the relative importance among sampled transitions by extracting local and global contexts from each of them and all sampled ones, respectively. In particular, NERS is designed to take a set of each experience's features as input and compute its outputs in an equivariant manner with respect to the permutation of the set. Here, we consider various features of transition such as TD-error, $Q$-value and the raw transition, e.g., expecting to sample intermediate transitions as those in blue boxes of Figure 1(c)) efficiently.

To verify the effectiveness of NERS, we validate the experience replay with various off-policy RL algorithms such as soft actor-critic (SAC) (Haarnoja et al., 2018a) and twin delayed deep deterministic (TD3) (Fujimoto et al., 2018) for continuous control tasks (Brockman et al., 2016; Todorov et al., 2012), and Rainbow (Hessel et al., 2018) for discontinuous control tasks (Bellemare et al., 2013). Our experimental results show that NERS consistently (and often significantly for complex tasks having high-dimensional state and action spaces) outperforms both the existing the rule-based (Schaul et al., 2016) and learning-based (Zha et al., 2019) sampling methods for experience replay.

In summary, our contribution is threefold:

- To the best of our knowledge, we first investigate the relative importance of sampled transitions for the efficient design of experience replays.

- For the purpose, we design a novel permutation-equivariant neural sampling architecture that utilizes contexts from the individual (local) and the collective (global) transitions with various features to sample not only meaningful but also diverse experiences.

- We validate the effectiveness of our neural experience replay on diverse continuous and discrete control tasks with various off-policy RL algorithms, on which it consistently outperforms both existing rule-based and learning-based sampling methods.

## 2 NEURAL EXPERIENCE REPLAY SAMPLER

We consider a standard reinforcement learning (RL) framework, where an agent interacts with an environment over discrete timesteps. Formally, at each timestep $t$, the agent receives a state $s_t$ from the environment and selects an action $a_t$ based on its policy $\pi$. Then, the environment returns a reward $r_t$, and the agent transitions to the next state $s_{t+1}$. The goal of the agent is to learn the policy $\pi$ that maximizes the return $R_t = \sum_{k=0}^{\infty} \gamma^k r_{t+k}$, which is the discounted cumulative reward from the timestep $t$ with a discount factor $\gamma \in [0, 1)$, at each state $s_t$. Throughout this section, we focus on off-policy actor-critic RL algorithms with a buffer $\mathcal{B}$, which consist of the policy $\pi_\psi(a|s)$ (i.e., actor) and $Q$-function $Q_\theta(s, a)$ (i.e., critic) with parameters $\psi$ and $\theta$, respectively.

### 2.1 OVERVIEW OF NERS

We propose a novel neural sampling policy $f$ with parameter $\phi$, called Neural Experience Replay Sampler (NERS). It is trained for learning to select important transitions from the experience replay buffer for maximizing the actual cumulative rewards. Specifically, at each timestep, NERS receives a set of off-policy transitions' features, which are proportionally sampled in the buffer $\mathcal{B}$ based on priorities evaluated in previous timesteps. Then it outputs a set of new scores from the set, in order for the priorities to be updated. Further, both the sampled transitions and scores are used to optimize the off-policy policy $\pi_\psi(a|s)$ and action-value function $Q_\theta(s, a)$. Note that the output of NERS should be equivariant of the permutation of the set, so we design its neural architecture to satisfy the property. Next, we define the reward $r^{\text{re}}$ as the actual performance gain, which is defined as the difference of the expectation of the sum of rewards between the current and previous evaluation policies, respectively. Figure 2 shows an overview of the proposed framework, which learns to sample from the experience replay. In the following section, we describe our method of learning the sampling policy for experience replay and the proposed network architecture in detail.

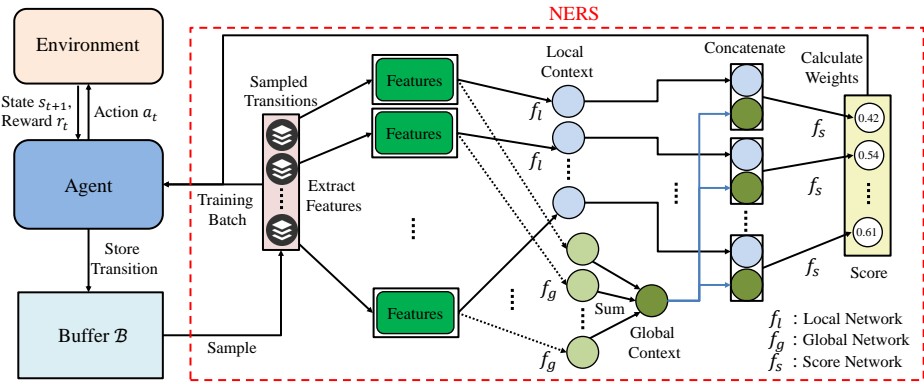

Figure 2: An overview of our neural experience replay sampler (NERS) framework. We first sample transitions proportionally to scores previously calculated. Then, our neural sampling policy evaluates them. Specifically, NERS consists of three networks $f_l$, $f_g$ and $f_s$. The first two networks obtain local and global contexts by considering various features, respectively. Then the last network evaluates the relative importance (score) by $f_s$. The importance set is used when to sample transitions later and train the agent. This design satisfies the permutation-equivariant property.

### 2.2 DETAILED COMPONENTS OF NERS

**Input observations.** Throughout this paper, we denote the set $\{1, \cdots, n\}$ by $[n]$ for positive integer $n$. Without loss of generality, suppose that the replay buffer $\mathcal{B}$ stores the following information as its $i$-th transition $\mathcal{B}_i = (s_{\kappa(i)}, a_{\kappa(i)}, r_{\kappa(i)}, s_{\kappa(i)+1})$ where $\kappa(i)$ is a function from the index of $\mathcal{B}$ to a corresponding timestep. We use a set of priorities $\mathcal{P}_\mathcal{B} = \{\sigma_1, \cdots, \sigma_{|\mathcal{B}|}\}$ that is updated whenever sampling transitions for training the actor and critic. One can sample an index set $I$ in $[|\mathcal{B}|]$ with the probability $p_i$ of $i$-th transition as follows:

$$p_i = \frac{\sigma_i^\alpha}{\sum_{k \in [|\mathcal{B}|]} \sigma_k^\alpha}, \tag{1}$$

---

**Algorithm 1** Training NERS: batch size $m$ and sample size $n$

---

Initialize NERS parameters $\phi$, a replay buffer $\mathcal{B} \leftarrow \emptyset$, priority set $\mathcal{P}_\mathcal{B} \leftarrow \emptyset$, and index set $\mathcal{I} \leftarrow \emptyset$
**for** each timestep $t$ **do**
    Choose $a_t$ from the actor and collect a sample $(s_t, a_t, r_t, s_{t+1})$ from the environment
    Update replay buffer $\mathcal{B} \leftarrow \mathcal{B} \cup \{(s_t, a_t, r_t, s_{t+1})\}$ and priority set $\mathcal{P}_\mathcal{B} \leftarrow \mathcal{P}_\mathcal{B} \cup \{1.0\}$
    **for** each gradient step **do**
        Sample an index $I$ by the given set $\mathcal{P}_\mathcal{B}$ and Eq. (1) with $|I| = m$
        Calculate a score set $\{\sigma_k\}_{k \in I}$ and weights $\{w_i\}_{i \in I}$ by Eq. (4) and Eq. (5), respectively
        Train the actor and critic using batch $\{\mathcal{B}_i\}_{i \in I} \subset \mathcal{B}$ and corresponding weights $\{w_i\}_{i \in I}$
        Collect $\mathcal{I} \leftarrow \mathcal{I} \bigcup I$ and update $\mathcal{P}_\mathcal{B}(I)$ by the score set $\{\sigma_k\}_{k \in I}$
    **end for**
    **for** the end of an episode **do**
        Choose a subset $I_\texttt{train}$ from $\mathcal{I}$ uniformly at random such that $|I_\texttt{train}| = n$
        Calculate $r^\texttt{re}$ as in Eq. (6)
        Update sampling policy $\phi$ using the gradient (7) with respect to $I_\texttt{train}$
        Empty $\mathcal{I}$, i.e., $\mathcal{I} \leftarrow \emptyset$
    **end for**
**end for**

---

with a hyper-parameter $\alpha > 0$. Then, we define the following sequence of features for $\{\mathcal{B}_i\}_{i \in I}$:

$$\mathbf{D}(\mathcal{B}, I) = \left\{ s_{\kappa(i)}, a_{\kappa(i)}, r_{\kappa(i)}, s_{\kappa(i)+1}, \kappa(i), \delta_{\kappa(i)}, r_{\kappa(i)} + \gamma \max_a Q_{\widehat{\theta}}\left(s_{\kappa(i)} + a\right) \right\}_{i \in I}, \quad (2)$$

where $\gamma$ is a discount factor, $\widehat{\theta}$ is the target network parameter, and $\delta_{\kappa(i)}$ is the TD-error defined as follows:

$$\delta_{\kappa(i)} = r_{\kappa(i)} + \gamma \max_a Q_{\widehat{\theta}}\left(s_{\kappa(i)+1}, a\right) - Q_\theta\left(s_{\kappa(i)}, a_{\kappa(i)}\right).$$

The TD-error indicates how 'surprising' or 'unexpected' the transition is (Schaul et al., 2016). Note that the input $\mathbf{D}(\mathcal{B}, I)$ contains various features including both exact values (i.e., states, actions, rewards, next states, and timesteps) and predicted values in the long-term perspective (i.e., TD-errors and $Q$-values). We abbreviate the notation $\mathbf{D}(\mathcal{B}, I) = \mathbf{D}(I)$ for simplicity. Utilizing various information is crucial in selecting diverse and important transitions (see Section 3).

**Architecture and action spaces.** Now we explain the neural network structure of NERS $f$. Basically, $f$ takes $\mathbf{D}(I)$ as an input and generate their scores, where these values are used to sample transitions proportionally. Specifically, $f$ consists of $f_l$, $f_g$, and $f_s$ called learnable local, global and score networks with output dimensions $d_l$, $d_g$, and 1. The local network is used to capture attributes in each transition by $f_l(\mathbf{D}(I)) = \{f_{l,1}(\mathbf{D}(I)), \cdots f_{l,|I|}(\mathbf{D}(I))\} \in \mathbb{R}^{|I| \times d_l}$, where $f_{l,k}(\mathbf{D}(I)) \in \mathbb{R}^{d_l}$ $(k \in [|I|])$. The global network is used to aggregate collective information of transitions by taking $f_g^\texttt{avg}(\mathbf{D}(I)) = \frac{\sum f_g(\mathbf{D}(I))}{|I|} \in \mathbb{R}^{1 \times d_g}$, where $f_g(\mathbf{D}(I)) \in \mathbb{R}^{|I| \times d_g}$. Then by concatenating them, one can make an input for the score network $f_s$ as follows:

$$\mathbf{D}^\texttt{cat}(I) := \left\{ f_{l,1}(\mathbf{D}(I)) \oplus f_g^\texttt{avg}(\mathbf{D}(I)), \cdots, f_{l,|I|}(\mathbf{D}(I)) \oplus f_g^\texttt{avg}(\mathbf{D}(I)) \right\} \in \mathbb{R}^{|I| \times (d_l + d_g)}, \quad (3)$$

where $\oplus$ denotes concatenation. Finally, the score network generates a score set:

$$f_s(\mathbf{D}^\texttt{cat}(I)) = \{\sigma_i\}_{i \in I} \in \mathbb{R}^{|I|}. \quad (4)$$

One can easily observe that $f_s$ is permutation-equivariant with respect to input $\mathbf{D}(I)$. The set $\{\sigma_i\}_{i \in I}$ is used to update the priorities set $\mathcal{P}$ for transitions corresponding to $I$ by Eq. (1) and to compute importance-sampling weights for updating the critic, compensating the bias of probabilities (Schaul et al., 2016)):

$$w_i = \left(\frac{1}{|\mathcal{B}|p(i)}\right)^\beta, \quad (5)$$

where $\beta > 0$ is a hyper-parameter. Then the agent and critic receive training batch $\mathbf{D}(I)$ and corresponding weights $\{w_i\}_{i \in I}$ for training, i.e., the learning rate for training sample $\mathcal{B}_i$ is set to be proportional to $w_i$. Due to this structure satisfying the permutation-equivariant property, one

can evaluate the relative importance of each transition by observing not only itself but also other transitions.

**Reward function and optimizing sampling policy.** We update NERS at each evaluation step. To optimize our sampling policy, we define the replay reward $r^{\mathtt{re}}$ of the current evaluation as follows: for policies $\pi$ and $\pi'$ used in the current and previous evaluations as in (Zha et al., 2019),

$$r^{\mathtt{re}} := \mathbb{E}_{\pi}\left[\sum_{t \in \{\text{timesteps in an episode}\}} r_t\right] - \mathbb{E}_{\pi'}\left[\sum_{t \in \{\text{timesteps in an episode}\}} r_t\right]. \tag{6}$$

The replay reward is interpreted as measuring how much actions of the sampling policy help the learning of the agent for each episode. Notice that $r^{\mathtt{re}}$ only observes the difference of the mean of cumulative rewards between the current and previous evaluation policies since NERS needs to choose transitions without knowing which samples will be added and how well agents will be trained in the future. To maximize the sample efficiency for learning the agent's policy, we propose to train the sampling policy to selects past transitions in order to maximize $r^{\mathtt{re}}$. To train NERS, one can choose $I_{\mathtt{train}}$ that is a subset of a index set $\mathcal{I}$ for totally sampled transitions in the current episode. Then we use the following formula by REINFORCE (Williams, 1992):

$$\nabla_{\phi}\mathbb{E}_{I_{\mathtt{train}}}\left[r^{\mathtt{re}}\right] = \mathbb{E}_{I_{\mathtt{train}}}\left[r^{\mathtt{re}}\sum_{i \in I_{\mathtt{train}}}\nabla_{\phi}\log p_i\left(\mathbf{D}\left(I_{\mathtt{train}}\right)\right)\right], \tag{7}$$

where $p_i$ is defined in Eq. (1). The detailed description is provided in Algorithm 1.

While ERO Zha et al. (2019) uses a similar replay-reward (Eq. 6), there are a number of fundamental differences between it and our method. First of all, ERO does not consider the relative importance between the transitions as NERS does, but rather learns an individual sampling rate for each transition. Moreover, they consider only three types of features, namely TD-error, reward, and the timestep, while NERS considers a larger set of features by considering more informative features that are not used by ERO, such as raw features, Q-values, and actions. However, the most important difference between the two is that ERO performs two-stage sampling, where they first sample with the individually learned Bernoulli sampling probability for each transition, and further perform random sampling from the subset of sampled transitions. However, with such a strategy, the first-stage sampling is highly inefficient even with moderate size experience replays, since it should compute the sampling rate for each individual instance. Accordingly, its time complexity of the first-stage sampling depends finally on the capacity of the buffer $\mathcal{B}$, i.e., $O\left(|\mathcal{B}|\right)$. On the contrary, NERS uses a sum-tree structure as in (Schaul et al., 2016) to sample transitions with priorities, so that its time complexity for sampling depends highly on $O\left(\log|\mathcal{B}|\right)$. Secondly, since the number of experiences selected from the first stage sampling is large, it may have little or no effect, making it to behave similarly to random sampling. Moreover, ERO updates its network with the replay reward and experiences that are not sampled from two-stage samplings but sampled by the uniform sampling at random (see Algorithm 2 in Zha et al. (2019)). In other words, samples that are never selected affect the training of ERO, while NERS updates its network solely based on the transitions that are actually selected by itself.

## 3 EXPERIMENTS

In this section, we conduct experiments to answer the following questions:

- Can the proposed sampling method improve the performances of various off-policy RL algorithms for both continuous and discrete control tasks?
- Is it really effective to sample diverse and meaningful samples by considering the relative importance with various contexts?

### 3.1 EXPERIMENTAL SETUP

**Environments.** In this section, we measure the performances of off-policy RL algorithms optimized with various sampling methods on the following standard continuous control environments

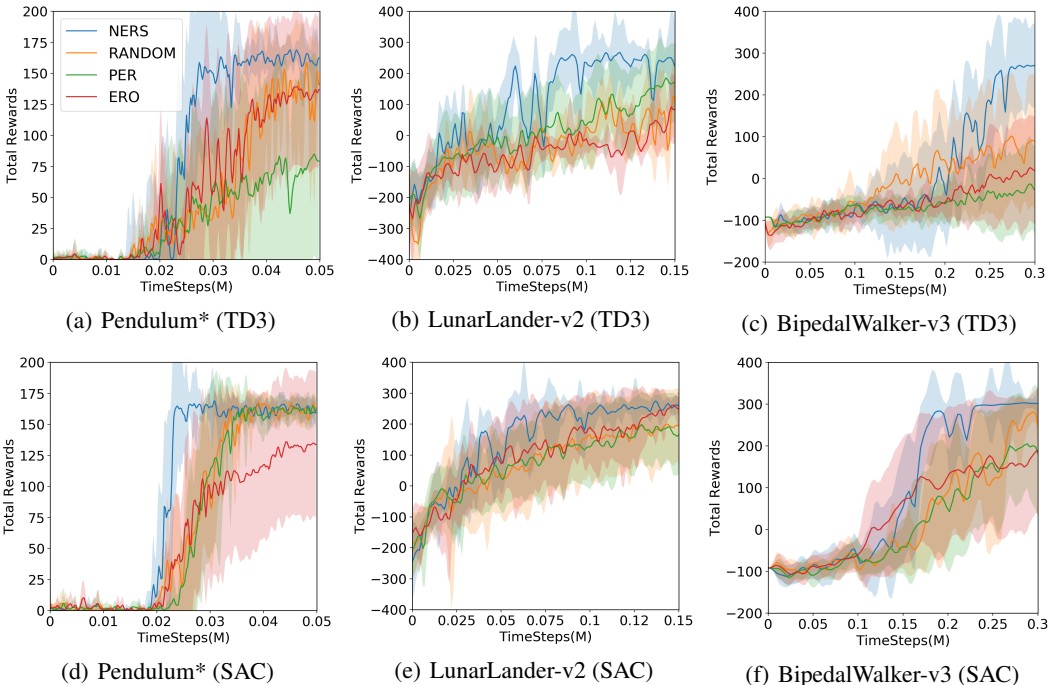

Figure 3: Learning curves of off-policy RL algorithms with various sampling methods on classical and Box2D continuous control tasks. The solid line and shaded regions represent the mean and standard deviation, respectively, across five runs with random seeds.

with simulated robots (e.g., Ant-v3, Walker2D-v3, and Hopper-v3) from the MuJoCo physics engine (Todorov et al., 2012) and classical and Box2D continuous control tasks (i.e., Pendulum*[2], LunarLanderContinuous-v2, and BipedalWalker-v3) from OpenAI Gym (Brockman et al., 2016). We also consider a subset of the Atari games (Bellemare et al., 2013) to validate the effect of our experience sampler on the discrete control tasks (see Table 2). The detailed description for environments is explained in supplementary material.

**Off-policy RL algorithms**. We apply our sampling policy to state-of-the-art off-policy RL algorithms, such as Twin delayed deep deterministic (TD3) (Fujimoto et al., 2018), and soft actor-critic (SAC) (Haarnoja et al., 2018a), for continuous control tasks. For discrete control tasks, instead of the canonical Rainbow (Hessel et al., 2018), we use a data-efficient variant of it as introduced in (van Hasselt et al., 2019). Notice that Rainbow already adopts PER. To compare sampling methods, we replaced it by NERS, RANDOM, and ERO in Rainbow, respectively. Due to space limitation, we provide more experimental details in the supplementary material.

**Baselines.** We compare our neural experience replay sampler (NERS) with the following baselines:

- RANDOM: Sampling transitions uniformly at random.
- PER (Prioritized Experience Replay): Rule-based sampling of the transitions with high temporal difference errors (TD-errors) (Schaul et al., 2016)
- ERO (Experience Replay Optimization): Learning-based sampling method (Zha et al., 2019), which computes the sampling score for each transition independently, using TD-error, timestep, and reward as features.

### 3.2 COMPARATIVE EVALUATION

Figure 3 shows learning curves of each off-policy RL algorithm during training on classical and Box2D continuous control tasks, respectively. Furthermore, Table 1 and Table 2 show the mean of cumulative rewards on MuJoCo and Atari environments after 500,000 and 100,000 training steps,

---

[2]Pendulum*: We slightly modify the original Pendulum that openAI Gym supports to distinguishing performances of sampling methods more clearly by making rewards sparser. Its detailed description is given in the supplementary material.

| Environments | NERS | RANDOM | PER | ERO |
|---|---|---|---|---|
| TD3 | | | | |
|     Ant-v3 | **4193.69 (+677.81)** | 2824.42 | 2723.57 | 3515.88 |
|     Walker2D-v3 | **3938.59 (+378.62)** | 3559.97 | 2797.86 | 3394.74 |
|     Hopper-v3 | **3062.32 (+451.92)** | 2152.82 | 1693.32 | 2610.40 |
| SAC | | | | |
|     Ant-v3 | **2913.54 (+678.20)** | 2235.34 | 1402.67 | 1844.99 |
|     Walker2D-v3 | **3720.11 (+323.12)** | 3035.31 | 3396.99 | 1057.61 |
|     Hopper-v3 | **2763.69 (+354.18)** | 2409.51 | 2223.08 | 2255.67 |

Table 1: Average of cumulative rewards under SAC and TD3 on MuJoCo Environments after 500,000 training steps across five instances with random seeds. Bold values represent the highest results, and the number in a bracket indicates the improvement due to NERS, compared to that of the best baseline on each environment.

| Environments | NERS | RANDOM | PER | ERO |
|---|---|---|---|---|
| Alien | **1125.18 (+281.16)** | 787.49 | 844.02 | 810.41 |
| Amidar | **167.50 (+18.85)** | 148.65 | 125.23 | 118.30 |
| Assualt | **516.52 (+26.46)** | 490.06 | 466.06 | 463.39 |
| Asterix | **679.00 (+80.90)** | 598.10 | 587.20 | 534.89 |
| BattleZone | **18584.99 (+1500.19)** | 14643.43 | 13870.98 | 17084.80 |
| Boxing | **2.51 (+4.83)** | -3.07 | -2.32 | -3.25 |
| ChopperCommand | **878.73 (+125.87)** | 696.34 | 752.86 | 727.45 |
| Freeway | **28.96 (+0.12)** | 28.09 | 28.37 | 28.84 |
| Frostbite | **1707.10 (+280.88)** | 794.50 | 1426.22 | 832.45 |
| KungFuMaster | **10925.59 (+2971.23)** | 7215.50 | 7527.96 | 7954.36 |
| MsPacman | **1579.27 (+432.45)** | 1070.19 | 1146.82 | 1001.87 |
| Pong | **-18.36 (+0.26)** | -18.62 | -19.08 | -18.76 |
| PrivateEye | **91.68 (+13.16)** | 69.84 | 78.52 | 56.13 |
| Qbert | **1037.58 (+136.34)** | 824.15 | 901.24 | 895.49 |
| RoadRunner | **9689.30 (+2596.17)** | 6382.82 | 7093.13 | 6199.88 |
| Seaquest | **386.80 (+30.52)** | 356.28 | 343.97 | 338.79 |

Table 2: Average of cumulative rewards under Rainbow on each Atari environments after 100,000 training steps across five instances. Bold values represent the highest results, and the number in a bracket indicates the improvement due to NERS, compared to that of the best baseline on each environment.

respectively, over five runs with random seeds, respectively.[3] We observe that NERS consistently outperforms baseline sampling methods in all tested cases. In particular, It significantly improves the performance of all off-policy RL algorithms on various tasks, which come with high-dimensional state and action spaces. These results imply that sampling good off-policy data is crucial in improving the performance of off-policy RL algorithms. Furthermore, they demonstrate the effectiveness of our method for both continuous and discrete control tasks, as it obtains significant performance gains on both types of tasks. On the other hand, we observe that PER, which is the rule-based sampling method, often shows worse performance than uniform random sampling (i.e., RANDOM) on these continuous control tasks, similarly as observed in (Zha et al., 2019). We suspect that this is because PER is more appropriate for $Q$-learning based algorithms than for policy-based learning, since TD-errors are used to update the $Q$ network. Moreover, even though ERO is a learning-based sampling method, its performance and sampling behavior is close to that of RANDOM, due to two reasons. First, it considers the importance of each transition individually by assuming the Bernoulli distribution, which may result in sampling of redundant transitions. Second, ERO performs two-stage sampling, where the transitions are first sampled due to their individual importance, and then further randomly sampled to construct a batch. However, since too many transitions are sampled in the first stage, the second-stage random sampling is similar to random sampling from the entire experience replay.

---

[3]Learning curves for each environment are provided in the supplementary material.

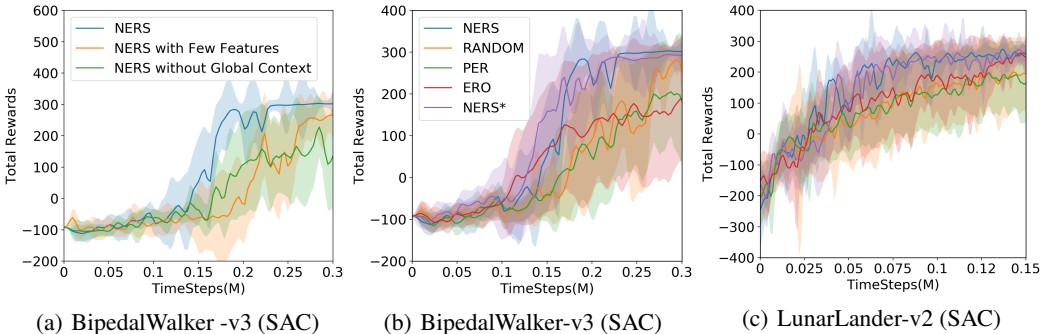

(a) BipedalWalker -v3 (SAC)   (b) BipedalWalker-v3 (SAC)   (c) LunarLander-v2 (SAC)

Figure 4: (a): Comparison of NERS and variants of NERS only with few features (reward, TD-error, and timestep) and without global context across five instances with random seeds, respectively. (b)-(c): Learning curves under SAC over five instances with random seeds across five instances with random seeds, respectively. Here, NERS* denotes that a variant of NERS, where it is trained by the difference of cumulative rewards from each training episode. Any significant difference between NERS and NERS* is not observable.

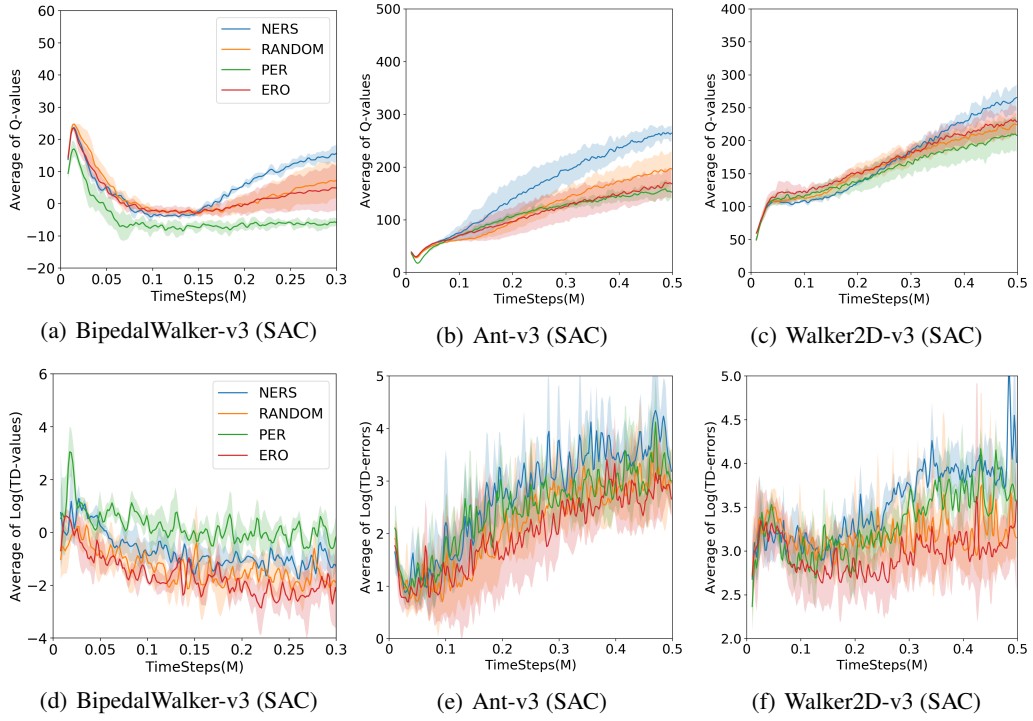

(a) BipedalWalker-v3 (SAC)   (b) Ant-v3 (SAC)   (c) Walker2D-v3 (SAC)

(d) BipedalWalker-v3 (SAC)   (e) Ant-v3 (SAC)   (f) Walker2D-v3 (SAC)

Figure 5: Curves of the average of $Q$-values (a/b/c) and Log(TD-errors) (d/e/f) of sampled transitions across five instances with random seeds, respectively. One can show that NERS basically selects transitions with high TD-errors in the beginning and both high TD-errors and Q-values finally.

### 3.3 ANALYSIS OF OUR FRAMEWORK

In this subsection, we first show that each component of NERS is crucial to improve sample efficiency (Figure 4). Next, we show that NERS really samples not only diverse but also meaningful transitions to update both actor and critic (Figure 5).

**Contribution by each component.** We analyze NERS to better understand the effect of each component. Figure 4(a) validates the contributions of our suggested techniques, where one can observe that the performance of NERS is significantly improved when using the full set of features. This implies that essential transitions for training can be sampled only by considering various aspects of the past experiences. Using only few features such as reward, TD-error, and timestep does not result in sampling transitions that yield high expected returns in the future. Figure 4(a) also shows the effect

| Method | STDEV of TD-errors | STDEV of $Q$-values | AVG of TD-errors | AVG of $Q$-values |
|--------|--------------------|--------------------|------------------|------------------|
| NERS | 723.01 | 65.22 | 87.54 | -104.13 |
| RANDOM | 528.76 | 60.46 | 62.43 | -120.15 |
| PER | 1256.71 | 49.78 | 139.49 | -138.05 |
| ERO | 560.56 | 59.16 | 65.44 | -119.03 |

Table 3: Sampled transitions' statistical values for $Q$-values and TD-errors on Pendulum-v0 under SAC at 10,000 training steps with initially 1,000 random actions. Here, STDEV and AVG mean the standard deviation and the average, respectively. PER has the highest STDEV of TD-errors but lowest STDEV of $Q$-errors. NERS has higher STDEV of both TD-errors and $Q$values than RANDOM and ERO.

of the relative importance by comparing NERS with and without considering the global context; we found that the sample efficiency is significantly improved due to consideration of the relative importance among sampled transitions, via learning the global context. Furthermore, although we have considered standard environments where evaluations are free, if there exists an environment where the total number of evaluations is restricted, it may be hard to calculate the replay reward in Eq.(6) since cumulative rewards at each evaluation should be computed. Due to this reason, we consider a variance of NERS (NERS*) which computes the difference of cumulative rewards in not evaluations but training episodes. Figure 4(b) and Figure 4(c) show the performance of NERS* compared to NERS and other sampling methods under BipedalWalker-v3 and LunearLanderContinuous-v2, respectively. These figures show that the performance between the two types of replay rewards is not significantly different.

**Analysis on statistics of sampled transitions.** We now check if NERS samples both meaningful and diverse transitions by examining how its sampling behavior changes during the training. To this end, we plot the TD-errors and Q-values for the sampled transitions during training on BipedalWalker-v3, Ant-v3, and Walker2D-v3 under SAC in Figure 5. We can observe that NERS learns to focus on sampling transitions with high TD-errors in the early training steps, while it samples transitions with both high TD-errors and $Q$-values (diverse) at later training iterations. In the early training steps, the critic network for value estimation may not be well trained, rendering the excessive learning of the agent to be harmful, and thus it is reasonable that NERS selects transitions with high TD-errors to focus on updating critic networks (Figure 5(d-f)), while it focuses both on transitions with both high Q-values and TD-errors since both the critic and the actor will be reliable in the later stage (Figure 5(a-c)). Such an adaptive sampling strategy is a unique trait of NERS that contributes to its success, while other sampling methods, such as PER and ERO, cannot do so. Table 3 denotes the statistical values for sampled transitions' TD-errors and $Q$-values on Pendulum-v3 under SAC at 10,000 steps (with initially 1,000 random actions). It is observable that NERS has higher standard deviation of $Q$-values and TD-errors than RANDOM and ERO. Although PER has the highest standard deviation of TD-errors than other sampling methods, it has the lowest standard deviation of $Q$-values instead. Figure 5 and Table 3 show that NERS learns to sample diverse, which means the NERS's ability to sample transitions with different criteria, and meaningful experiences for agents.

## 4 RELATED WORK

**Off-policy algorithms.** One of the well-known off-policy algorithms is deep $Q$-network (DQN) learning with a replay buffer (Mnih et al., 2015). There are various variants of the DQN learning, e.g., (Hasselt, 2010; Wang et al., 2015; Hessel et al., 2018). Especially, Rainbow (Hessel et al., 2018), which is one of the state-of-the-art $Q$-learning algorithms, was proposed by combining various techniques to extend the original DQN learning. Moreover, DQN was combined with a policy-based learning, so that various actor-critic algorithms have appeared. For instance, an actor-critic algorithm, which is called deep deterministic policy gradient (DDPG) (Lillicrap et al., 2016), specialized for continuous control tasks was proposed by a combination of DPG (Silver et al., 2014) and deep Q-learning (Mnih et al., 2015). Since DDPG is easy to brittle for hyper-parameters setting, various algorithms have been proposed to overcome this issue. For instance, to reduce the the overestimation of the $Q$-value in DDPG, twin delayed DDPG (TD3) was proposed (Fujimoto et al., 2018), which extended DDPG by applying double $Q$-networks, target policy smoothing, and different frequencies

to update a policy and $Q$-networks, respectively. Moreover, another actor-critic algorithm called soft actor-critic (SAC) (Haarnoja et al., 2018a;b) was developed by adding the entropy measure of an agent policy to the reward in the actor-critic algorithm to encourage the exploration of the agent.

**Sampling method.** Due to the ease of applying random sampling, it has been used to various off-policy algorithms until now. However, it is known that it cannot guarantee optimal results, so that a prioritized experience replay (PER) (Schaul et al., 2016) that samples transitions proportionally to the TD-error in DQN learning was proposed. As a result, it showed performance improvements in Atari environments. Applying PER is also easily applicable to various policy-based algorithms, so it is one of the most frequently used rule-based sampling methods (Hessel et al., 2018; Hou et al., 2017; Schaul et al., 2016; Wang & Ross, 2019). Furthermore, since it is reported that the newest experiences are significant for efficient $Q$-learning (Zhang & Sutton, 2015), PER generally imposes the maximum priority on recent transitions to sample them frequently. Based on PER, imposing weights for recent transitions was also suggested (Brittain et al., 2019) to increase priorities for them. Instead of TD-error, a different metric can be also used to PER, e.g., the expected return (Isele & Cosgun, 2018; Jaderberg et al., 2016). Meanwhile, different approaches from PER have been proposed. For instance, to update the policy in a trust region, computing the importance weight of each transitions was proposed (Novati & Koumoutsakos, 2019), so far-policy experiences were ignored when computing the gradient. Another example is backward updating of transitions from a whole episode (Lee et al., 2019) for deep $Q$-learning. Although the rule-based methods have shown their effectiveness on some tasks, they sometimes derive sub-optimal results on other tasks. To overcome this issue, a neural network for replay buffer sampling was adopted (Zha et al., 2019) and it showed the validness of their method on some continuous control tasks in the DDPG algorithm. However, its effectiveness is arguable in other tasks and algorithms (see Section 3), as it only considers transitions independently and regard few features as timesteps, rewards, and TD-errors (unlike ours). Recently, Fedus et al. (2020) showed that increasing replay capacity and downweighting the oldest transition in the buffer generally improves the performance of $Q$-learning agents on Atari tasks. How to sample prior experiences is also a crucial issue to model-based RL algorithms, e.g., Dyna Sutton (1991) which is a classical architecture. There are variants of Dyna that study strategies for search-control, to selects which states to simulate. For instance, inspired by the fact that a high-frequency space requires many samples to learn, Dyna-Value Pan et al. (2019) and Dyna-Frequency Pan et al. (2020) select states with high-frequency hill climbing on value function, and gradient and hessian norm of it, respectively for generating more samples by the models. In other words, how to prioritize transitions when sampling is nontrivial, and learning the optimal sampling strategy is critical for the sample-efficiency of the target off-policy algorithm.

## 5  CONCLUSION

We proposed NERS, a neural policy network that learns how to select transitions in the replay buffer to maximize the return of the agent. It predicts the importance of each transition in relation to others in the memory, while utilizing local and global contexts from various features in the sampled transitions as inputs. We experimentally validate NERS on benchmark tasks for continuous and discrete control with various off-policy RL methods, whose results show that it significantly improves the performance of existing off-policy algorithms, with significant gains over prior rule-based and learning-based sampling policies. We further show through ablation studies that this success is indeed due to modeling relative importance with consideration of local and global contexts.

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

# Supplementary Material:

# Learning to Sample with Local and Global Contexts in Experience Replay Buffer

## A    ENVIRONMENT DESCRIPTION

### A.1    MUJOCO ENVIRONMENTS

Multi-Joint Dynamics with Contact (MuJoCo) Todorov et al. (2012) is a physics engine for robot simulations supported by openAI gym[4]. MuJoCo environments provide a robot with multiple joints and reinforcement learning (RL) agents should control the joints (action) to achieve a given goal. The observation of each environment basically includes information about the angular velocity and position for those joints. In this paper, we consider the following environments belonging to MuJoCo.

**Hopper(-v3)** is a environment to control a one-legged robot. The robot receives a high return if it hops forward as soon as possible without failure.

**Walker2d(-v3)** is an environment to make a two-dimensional bipedal legs to walk. Learning to quick walking without failure ensures a high return.

**Ant(-v3)** is an environment to control a creature robot with four legs used to move. RL agents should to learn how to use four legs for moving forward quickly to get a high return.

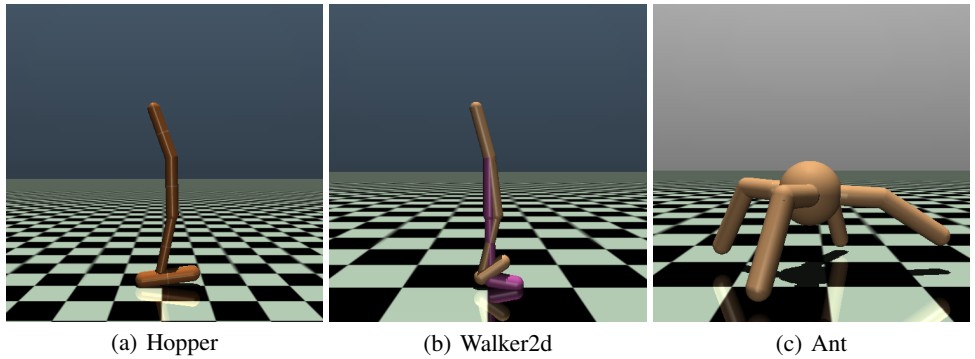

|  (a) Hopper | (b) Walker2d | (c) Ant |

Figure A.1: MuJoCo environments

### A.2    OTHER CONTINUOUS CONTROL ENVIRONMENTS

Although MuJoCo environments are popular to evaluate RL algorithms, openAI gym also supports additional continuous control environments which belong to classic or Box2D simulators. We conduct experiments on the following environments among them.

**Pendulum*** is an environment which objective is to balance a pendulum in the upright position to get a high return. Each observation represents the angle and angular velocity. An action is a joint effort which range is $[-2, 2]$. Pendulum* is slightly modified from the original (Pendulum-v0) which openAI supports. The only difference from the original is that agents receive a reward $1.0$ only if the rod is in sufficiently upright position (between the angle in $[-\pi/3, \pi/3]$, where the zero angle means that the rod is in completely upright position) at least more than $20$ steps.

**LunarLander(Continuous-v2)** is an environment to control a lander. The objective of the lander is landing to a pad, which is located at coordinates $(0, 0)$, with safety and coming to rest as soon as possible. There is a penalty if the lander crashes or goes out of the screen. An action is about parameters to control engines of the lander.

---

[4]https://gym.openai.com/

**BipedalWalker(-v3)** is an environment to control a robot. The objective is to make the robot move forward far from the initial state as far as possible. An observation is information about hull angle speed, angular velocity, vertical speed, horizontal speed, and so on. An action consists of torque or velocity control for two hips and two knees.

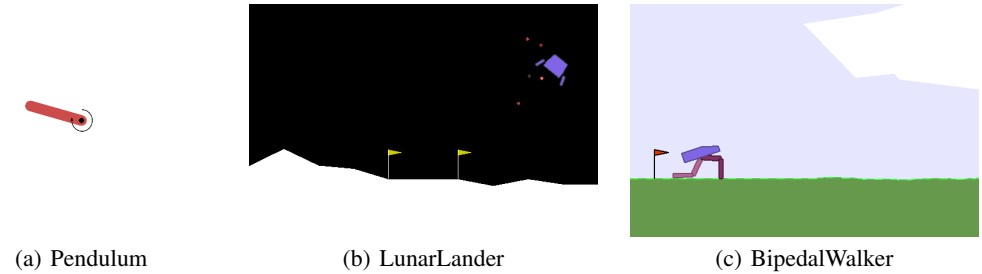

(a) Pendulum        (b) LunarLander        (c) BipedalWalker

Figure A.2: Other continuous control environments

Table A.1 describes the observation and action spaces and the maximum steps for each episode (horizon) in MuJoCo and other continuous control environments. Here, $\mathbb{R}$ and $[-1, 1]$ denote sets of real numbers and those between $0$ and $1$, respectively.

| Environment | Observation space | Action space | Horizon |
|---|---|---|---|
| Hopper | $\mathbb{R}^{11}$ | $[-1,1]^3$ | 1000 |
| Walker2d | $\mathbb{R}^{17}$ | $[-1,1]^6$ | 1000 |
| Ant | $\mathbb{R}^{111}$ | $[-1,1]^8$ | 1000 |
| Pendulum* | $\mathbb{R}^3$ | $[-1,1]^1$ | 200 |
| LunarLander | $\mathbb{R}^8$ | $[-1,1]^2$ | 1000 |
| BipedalWalker | $\mathbb{R}^{24}$ | $[-1,1]^4$ | 1600 |

Table A.1: Dimensions of observation and action spaces for continuous control environments

### A.3 DISCRETE CONTROL ENVIRONMENT

To evaluate sampling methods under Rainbow Hessel et al. (2018), we consider the following Atari environments. RL agents should learn their policy by observing the RGB screen to acheive high scores for each game.

**Alien(NoFrameskip-v4)** is a game where player should destroy all alien eggs in the RGB screen with escaping three aliens. The player has a weapon which paralyzes aliens.

**Amidar(NoFrameskip-v4)** is a game which format is similar to MsPacman. RL agents control a monkey in a fixed rectilinear lattice to eat pellets as much as possible while avoiding chasing masters. The monkey loses one life if it contacts with monsters. The agents can go to the next stage by visiting a certain location in the screen.

**Assault(NoFrameskip-v4)** is a game where RL agents control a spaceship. The spaceship is able to move on the bottom of the screen and shoot motherships which deploy smaller ships to attack the agents. The objective is to eliminate the enemies.

**Asterix(NoFrameskip-v4)** is a game to control a tornado. The objective of RL agents is to eat hamburgers in the screen with avoiding dynamites.

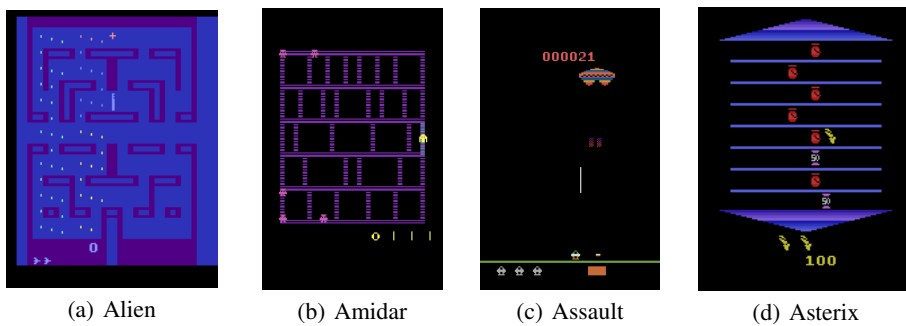

|  |  |  |  |
|:---:|:---:|:---:|:---:|
| (a) Alien | (b) Amidar | (c) Assault | (d) Asterix |

**BattleZone(NoFrameskip-v4)** is a tank combat game. This game provides a first-person perspective view. RL agents control a tank to destroy other tanks. The agent should avoid other tanks or missile attacks. It is also possible to hide from various obstacles and avoid enemy attacks.

**Boxing(NoFrameskip-v4)** is a game about the sport of boxing. There are two boxers with a top-down view and RL agents should control one of them. They get one point if their punches hit from a long distance and two points if their punches hit from a close range. A match is finished after two minues or 100 punches hitted to the opponent.

**ChopperCommand(NoFrameskip-v4)** is a game to control a helicopter in a desert. The helicopter should destroy all enemy aircrafts and helicopters while protecting a convoy of trucks.

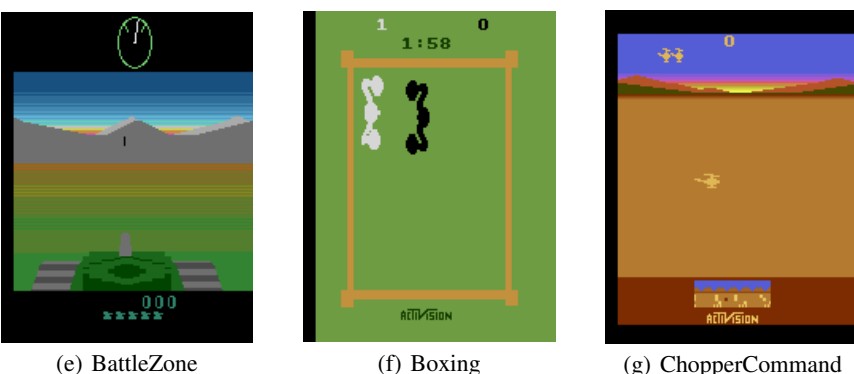

|  |  |  |
|:---:|:---:|:---:|
| (e) BattleZone | (f) Boxing | (g) ChopperCommand |

**Freeway(NoFrameskip-v4)** is a game where RL agents control chickens to run across a ten-lane highway with traffic. They are only allowed to move up or down. The objective is to get across as possible as they can until two minutes.

**Frostbite(NoFrameskip-v4)** is a game to control a man who should collect ice blocks to make his igloo. The bottom two thirds of the screen consists of four rows of horizontal ice blocks. He can move from the current row to another and obtain an ice block by jumping. RL agents are required to collect 15 ice blocks while avoiding some opponents, e.g., crabs and birds.

**KungFuMaster(NoFrameskip-v4)** is a game to control a fighter to save his girl friend. He can use two types of attacks (punch and kick) and move/crunch/jump actions.

**MsPacman(NoFrameskip-v4)** is a game where RL agents control a pacman in given mazes for eatting pellets as much as possible while avoiding chasing masters. The pacman loses one life if it contacts with monsters.

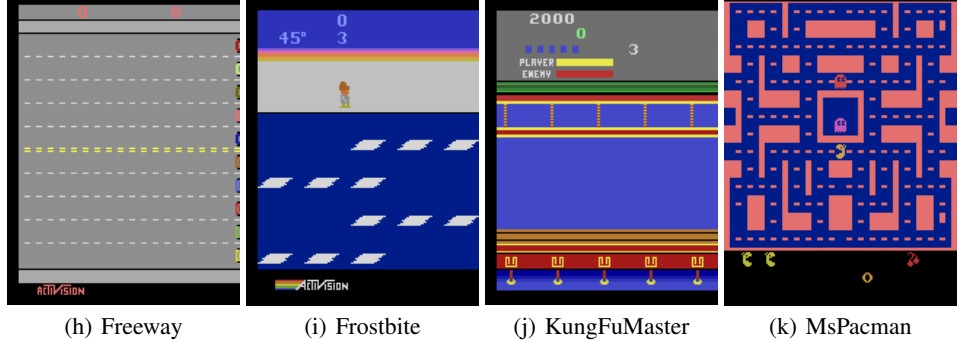

| (h) Freeway | (i) Frostbite | (j) KungFuMaster | (k) MsPacman |

**Pong(NoFrameskip-v4)** is a game about table tennis. RL agents control an in-game paddle to hit a ball back and forth. The objective is to gain 11 points before the opponent. The agents earn each point when the opponent fails to return the ball.

**PrivateEye(NoFrameskip-v4)** is a game mixing action, adventure, and memorizationm which control a private eye. To solve five cases, the private eye should find and return items to suitable places.

**Qbert(NoFrameskip-v4)** is a game where RL agents control a character under a pyramid made of 28 cubes. The character should change the color of all cubes while avoiding obstacles and enemies.

**RoadRunner(NoFrameskip-v4)** is a game to control a roadrunner (chaparral bird). The roadrunner runs to the left on the road. RL agents should pick up bird seeds while avoiding a chasing coyote and obstacles such as cars.

**Seaquest(NoFrameskip-v4)** is a game to control a submarine to rescue divers. It can also attack enemies by missiles.

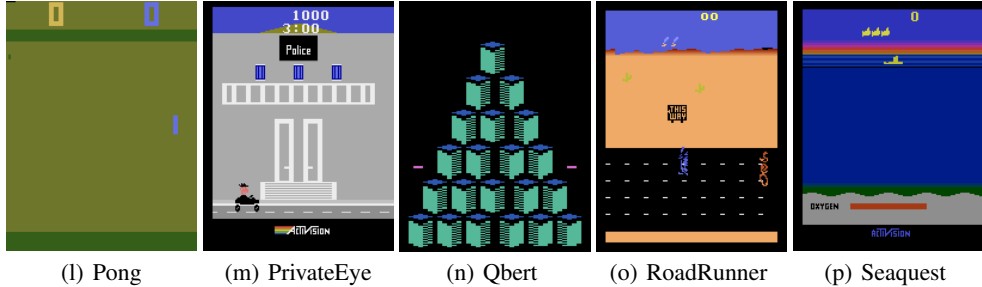

| (l) Pong | (m) PrivateEye | (n) Qbert | (o) RoadRunner | (p) Seaquest |

# B  TRAINING DETAILS

| Parameter | Value |
|---|---|
| **Shared** | |
| Batch size (continuous control environments) | 128 |
| Batch size (discrete control environments) | 32 |
| Buffer size | $10^6$ |
| Target smoothing coefficient ($\tau$) for soft update | $5 \times 10^{-3}$ |
| Initial prioritized experience replay buffer exponents $(\alpha, \beta)$ [5] | $(0.5, 0.4)$ |
| Discount factor for the agent reward ($\gamma$) | 0.99 |
| Number of initial random actions (continuous control environments) | $5 \times 10^3$ |
| Number of initial random actions (discrete control environments) | $1,600$ |
| Optimizer | Adam Kingma & Ba (2014) |
| Nonlinearity | ReLU |
| Observation down-sampling for Atari RGB | $84 \times 84$ with grey-scaling |
| CNN channels for Atari environments | 32, 64 |
| CNN filter size for Atari environments | $5 \times 5, 5 \times 5$ |
| CNN stride for Atari environments | 5, 5 |
| **ERO** | |
| Hidden units per layer | 64, 64 |
| Learning rate | $10^{-4}$ |
| **NERS** | |
| Hidden units per layer after flattening the output from CNNs | 256, 64, 32 |
| Hidden units per layer in the local and global networks ($f_l$ and $f_g$) | 256, 512, 256, 128, |
| Hidden units per layer in the score network ($f_s$) | 256, 128, 64 |
| Sampling size to update NERS ($n$) | 128 |
| Learning rate | $10^{-4}$ |
| **TD3** | |
| Hidden units per layer | 256, 256 |
| Learning rate | $5 \times 10^{-3}$ |
| Policy update frequency | 2 |
| Gaussian action and target noises | 0.1, 0.2 |
| Target noise clip | 0.5 |
| Target network update | Soft update with interval 1 |
| **SAC** | |
| Hidden units per layer | 256, 256 |
| Learning rate | $3 \times 10^{-4}$ |
| Target entropy | $-\dim A$ ($A$ is action space) |
| Target network update | Soft update with interval 1 |
| **Rainbow** | |
| Action repetitions and Frame stack | 4 |
| Reward clipping | True ($[-1, 1]$) |
| Terminal on loss of life | True |
| Max frames per episode | $1.08 \times 10^5$ |
| Target network update | Hard update (every 2,000 updates) |
| Support of $Q$-distribution | 51 |
| $\epsilon$ for Adam optimizer | $1.5 \times 10^4$ |
| Learning Rate | $10^{-4}$ |
| Max gradient norm | 10 |
| Noisy nets parameter | 0.1 |
| Replay period every | 1 |
| Multi-step return length | 20 |
| $Q$-network's hidden units per layer | 256 |

Table B.1: Hyper-parameters

---

[5]$\beta$ increases to 1.0 by the rule $\beta = 0.4\eta + 1.0(1 - \eta)$, where $\eta =$ the current step/the maximum steps.

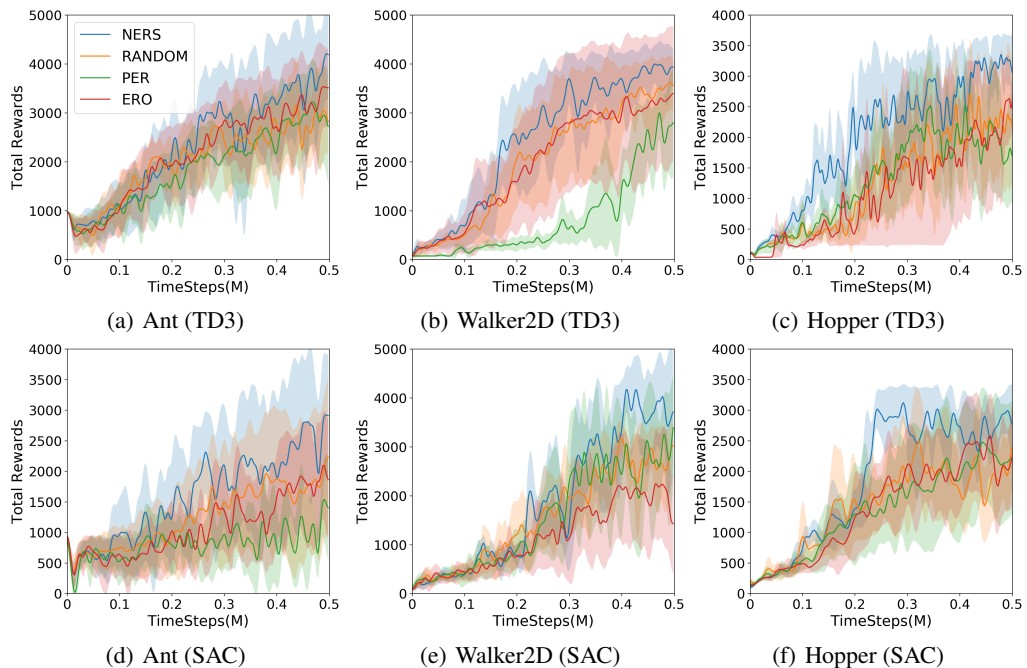

Figure C.1: Learning curves of off-policy RL algorithms with various sampling methods on MuJoCo tasks. The solid line and shaded regions represent the mean and standard deviation, respectively, across five instances.

Table B provides hyper-parameters which we used. We basically adopt parameters for Twin delayed DDPG (TD3) Fujimoto et al. (2018) and Soft actor critic (SAC) Haarnoja et al. (2018a;b) in openAI baselines [6]. Furthermore, we adopt parameters Rainbow as in van Hasselt et al. (2019) to make data efficient Rainbow for Atari environments. In the case of continuous control environments, we train five instances of TD3 and SAC, where they perform one evaluation rollout per the maximum steps. In the case of discrete control environments, we trained five instances of Rainbow, where they perform 10 evaluation rollouts per 1000 steps. During evaluations, we collect cumulative rewards to compute the replay reward $r^{\mathtt{re}}$.

We follow the hyper-parameters in van Hasselt et al. (2019) for prioritized experience replay (PER). We also use the hyper-parameters for experience replay optimization (ERO) used in Zha et al. (2019). Since NERS is interpreted as an extension of PER, it basically shares hyper-parameters in PER, e.g., $\alpha$ and $\beta$. NERS uses various features, e.g., TD-errors and $Q$-values, but the newest samples have unknown $Q$-values and TD-errors before sampling them to update agents policy. Accordingly, we normalize $Q$-values and TD-errors by taking the hyperbolic tangent function and set $1.0$ for the newest samples' TD-errors and $Q$-values. Furthermore, notice that NERS uses both current and next states in a transition as features, so that we adopt CNN-layers in NERS for Atari environments as in van Hasselt et al. (2019). After flattening and reducing the output of the CNN-layers by FC-layers (256-64-32) , we make a vector by concatenating the reduced output with the other features. Then the vector is input of both local and global networks $f_l$ and $f_g$. In the case of ERO, it does not use states as features, so that CNN-layers are unneccesary.

Our objective is not to achieve maximal performance but compare sampling methods. Accordingly, to evaluate sampling methods on Atari environments, we conduct experiments until 100,000 steps as in van Hasselt et al. (2019) although there is room for better performance if more learning. In the case of continuous control environments, we conduct experiments until 500,000 steps.

## C  ADDITIONAL EXPERIMENTAL RESULTS

Figure C.1 shows additional continuous control environments: Ant, Walker2d, and Hopper under TD3 and SAC, respectively. All tasks possess have high-dimensional observation and action spaces (see Table A.1). One can show that NERS outperforms other sampling methods at most cases. Moreover, one can observe that RANDOM and ERO have almost similar performance and PER could not show

---
[6]https://github.com/openai/baselines

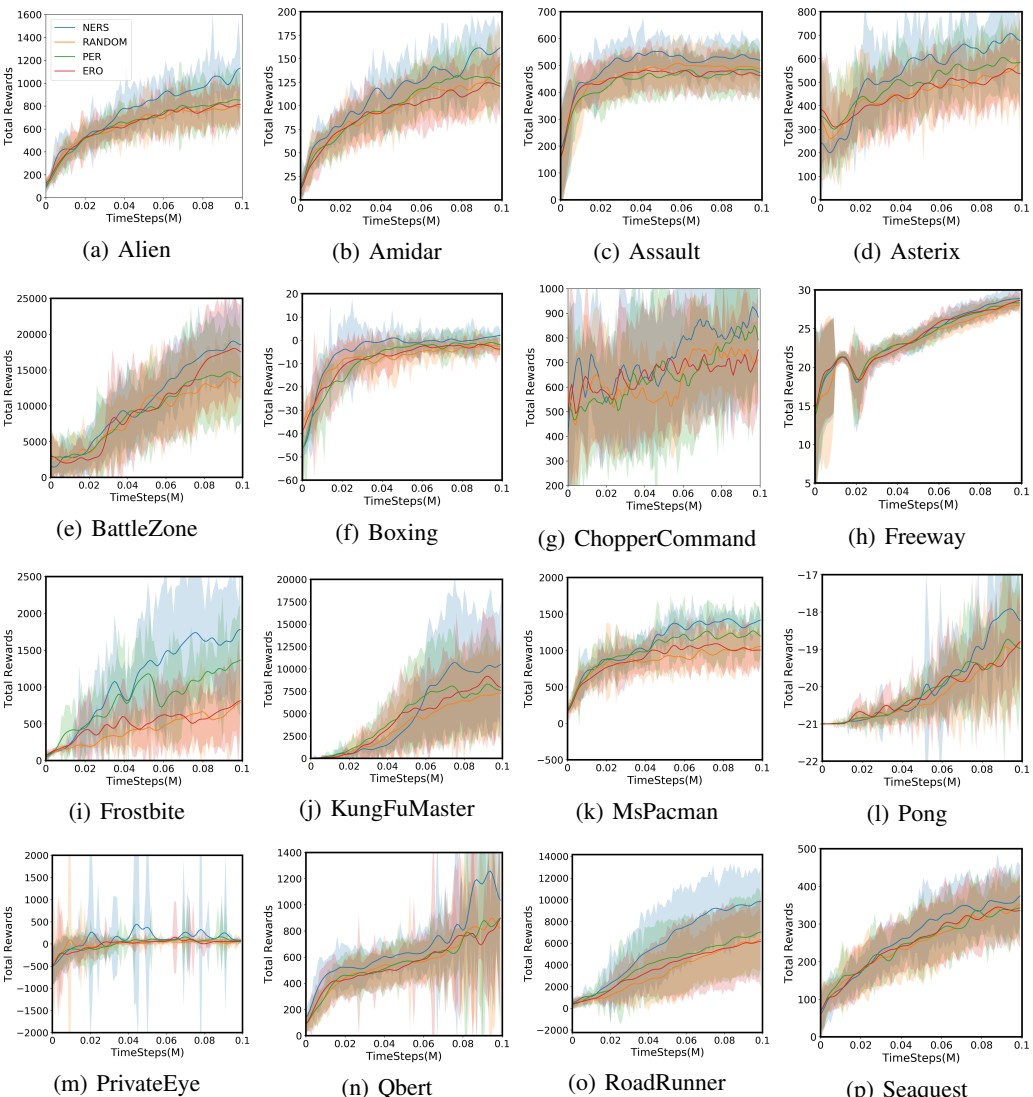

Figure C.2: Learning curves on additional Atari environments under Rainbow

better performance to policy-based RL algorithms compared to other sampling methods. Detailed learning curves of Rainbow for each environment are observable in Figure C.2.

We believe that in spite of the effectiveness of PER under Rainbow, the poor performance of PER under policy-based RL algorithms results from that it is specialized to update $Q$-newtorks, so that the actor networks cannot be efficiently trained.

One can observe that there are high variances in some environments. Indeed, it is known that learning more about environments in Figure C.1 and Figure C.2 improves performance of algorithms. However, our focus is not to obtain the high performance but to compare the speed of learning according to the sampling methods under the same off-policy algorithms, so we will not spend more timesteps.

