# OpenReview forum: "Learning to Sample with Local and Global Contexts  in Experience Replay Buffer"
_ICLR.cc/2021/Conference — ICLR 2021 Poster_

### Official Review · AnonReviewer3 · 2020-10-28
**Interesting idea after clarifying the misunderstanding.**

**Rating:** 6
**Confidence:** 3

**Review:**

##########################################################################

Summary:
This paper proposes to improve sampling from the experience replay buffers that weights samples by their "relatively usefulness".

This paper proposes to use two encoders - one global that encodes across the current batch of experience replay sample, and one local that encodes each selected timestep. Using the encodings, a scorer scores the experiences and weights the actor and critic losses in proportion to that score. The sampler is trained to maximize the probability that it chooses high reward timesteps on average.



##########################################################################

Pros:

Experience Replay is a widely used technique and improving on the naive random sample method makes sense. The analysis on other sampling methods is insightful.
The idea to have a global feature pooled from the sample transitions is unique.
Reweighing each sample's loss at training time is simple and effective.

##########################################################################

Cons:

The paper's idea could be flawed. The sampler is trained so that it prioritizes high reward timesteps as in eq 6.
But this is dubious. What if we need to sample the N timesteps right before a high-reward timestep, even though those preceding timesteps do not have high return themselves? And how do you explain why"NERS focuses on sampling transitions with high TD-errors in the beginning, ... as the timestep progresses, it samples transitions with both high TD-errors and Q-values (diverse)", given that it's trained with a single objective to maximize sampled reward? "using various features in an advanced manner." is not a satisfactory explaination.


The experiment section is not convincing.
The model has not been trained till convergence. e.g. SAC and ERO has all been trained with 1e6+ steps at least. In addition, what is the reason to do Hopper from Mujoco instead of Hopper-V2 from the OpenAI gym? With the latter, you can compare with the numbers published in ERO.

##########################################################################

Questions during rebuttal period:
Please address and clarify the cons above

---

> ### Author Response · Authors · 2020-11-15
> **Response to Reviewer #3**
>
> We thank you for your time and effort in reviewing our paper. We respond to your comments below:
>
> ---
> **Q1. The sampler is trained so that it prioritizes high reward timesteps as in eq 6. But this is dubious. What if we need to sample the N timesteps right before a high-reward timestep, even though those preceding timesteps do not have high return themselves?**
>
> - This is a critical misunderstanding. Please note that our NERS **does not**prioritize high-reward timesteps. Please note that Eq 6. is the expected **cumulative reward**, and not a reward at a specific timestep. Thus, our method will sample transitions before a high-reward timestep, if they help the model to eventually transit to the high-reward timestep, since doing so will maximize the **cumulative reward.**
>
> - We have shown this point with the experiments in the Pendulum* environment, which is modified from the original Pendulum environment to have **sparse rewards**, such that the agent receives a reward only when the rod is in a **near-upright position** (Please see the footnote in Page 5, and the description of Pendulum* environment in Page 13). Thus we need preceding transitions in order to get to the high-reward (upright) position, although they provide **no reward**at all. The Figure 3 (a) and (d) on page 6 shows that NERS achieves significantly better convergence and performance over baseline sampling methods in this sparse reward environment.
>
> ---
> **Q2. How do you explain why "NERS focuses on sampling transitions with high TD-errors in the beginning, ... as the timestep progresses, it samples transitions with both high TD-errors and Q-values (diverse)", given that it's trained with a single objective to maximize sampled reward? "using various features in an advanced manner." is not a satisfactory explanation.**
>
> - NERS is **not**trained to sample transitions with maximum rewards, and is trained to maximize the **expected cumulative rewards**, thus it will learn to sample any transitions that help maximize the **cumulative reward**, by dynamically weighting various features (e.g. TD-errors, Q-values, and raw features) based on their contributions in the course of training.
>
> - The plots of Q-value and TD-errors in Figure 5 thus shows that NERS is able to sample transitions by **dynamically focusing on different features**at different stages of training.  In the beginning, the critic network for value estimation is not well trained, and thus excessive learning of the agent may be harmful in early steps, and thus it is reasonable that NERS selects transitions with high TD-errors to focus on updating critic networks in early training iterations (Figure 5(d-f)), while focusing both on transitions with both high $Q$-values and TD-errors as training goes on (Figure 5(a-c)). This is a unique trait of NERS that contributes to its success.
>
> ---
>
> **Q3. The experiment section is not convincing. The model has not been trained till convergence. e.g. SAC and ERO has all been trained with 1e6+ steps at least.**
>
> - We report our results at 0.5M iteration, since many existing works that focus on sample-efficient reinforcement learning also report their results before full convergence [1, 2, 3, 4].
>
> - We believe that 0.5M is more than sufficient in our case, since we use SAC and TD3, which are known to converge faster than DDPG that is used by ERO.
>
> - Also, we empirically observed that in the MuJuCo environments, if there is are meaningful differences between different sampling methods in early steps (say, 0.3M steps), there is no change of their relative rankings even with more iterations. We will include the results with more number of iterations in the final version of the paper.
>
> ---
> **Q4. In addition, what is the reason to do Hopper from Mujoco instead of Hopper-V2 from the OpenAI gym? With the latter, you can compare with the numbers published in ERO.**
>
> - We apologize for the confusion. The MuJoCo tasks are completely the same as the tasks provided by the OpenAI gym. We have clarified the precise version information in all Figures in page 6, 7, and 8, in the revised manuscript.
> - Since the experiments in ERO [2] is conducted only with DDPG algorithm, it is difficult to compare with the reported performances. Note that we validated all sampling methods with two RL algorithms that are known to work better (SAC, TD3).
>
> ---
>
> [1] Fujimoto, S., H. van Hoof, and D. Meger. "Addressing function approximation error in actor-critic methods." Proceedings of Machine Learning Research 80 (2018): 1587-1596.
>
> [2] Wang, Che, and Keith Ross. "Boosting Soft Actor-Critic: Emphasizing Recent Experience without Forgetting the Past." arXiv preprint arXiv:1906.04009 (2019).
>
> [3] Zha, Daochen, et al. "Experience replay optimization." In Proceedings of the 28th International Joint Conference on Artificial Intelligence. AAAI Press, 2019. p. 4243-4249.
>
> [4] Haarnoja, Tuomas, et al. "Soft Actor-Critic Algorithms and Applications." arXiv (2018): arXiv-1812.

---

> > ### Comment · AnonReviewer3 · 2020-11-22
> > **Revision: Interesting idea after clarifying the misunderstanding.**
> >
> > > Q1. The sampler is trained so that it prioritizes high reward timesteps as in eq 6. But this is dubious. What if we need to sample the N timesteps right before a high-reward timestep, even though those preceding timesteps do not have high return themselves?
> >
> > * Thank you for pointing out my misunderstanding. The idea makes a lot more sense now it’s clearer to me what the sampler update part is actually doing. The other two reviewers shared the same feeling that it was unclear in the first version, especially around whether you get the expected reward of the current and past policy. I will revise my rating accordingly.
> >
> > > Q3. The experiment section is not convincing. The model has not been trained till convergence. e.g. SAC and ERO has all been trained with 1e6+ steps at least.
> >
> > * My comment “The model has not been trained till convergence.” came from Figure 3 rather than from Table 1, which I now see that you mentioned it was trained for 0.5M steps. Apologies for the seemingly rushed review.

---

### Official Review · AnonReviewer2 · 2020-10-29
**The paper proposes an interesting idea to design sampling distribution to improve sample efficiency of deep RL algorithms. But there are several nontrivial issues to be resolved.**

**Rating:** 6
**Confidence:** 4

**Review:**

Observing that the existed ER-based sampling method may introduce bias or redundancy in sampled transitions, the paper proposes a new sampling method in the ER learning setting. The idea is to take into consideration the context, i.e. many visited transitions, rather than a single one, based on which one can measure the relative importance of each transition. Specifically, the weights of transitions are also learned through a Reinforce agent and hence the sampling distribution is learned to directly improve sample efficiency.

Clarity. The presentation is clear in most places. But I do feel the core part of updating the sampling policy needs clarification.

Quality. Please see the questions below.

Originality/Significance. The method is novel and is potentially interesting to the RL research community.

In the original prioritized ER paper, they use a ratio to mitigate the biased sampling issue, did the authors ever visualize what the result would be (say in Figure 1) if you use that ratio?

Eq (2) the input feature can be highly non-stationary/unstable. For example, some of the variables may decrease all the time, and some others may increase all the time. Intuitively, training with such data should be very challenging. It looks like that in order to resolve the issue of biased sampling, the authors introduce an even more difficult task. Do the authors have some comments about this?

I am confused about how the sampling network is updated. In Algorithm 1, my understanding is that if the current time step is the end of an episode, then update the sampling network. Is it correct? Note that Algorithm 1 indicates that the actor, critic are updated at each time step. But the paper also says that the NERS is updated at each evaluation step and this means that throughout the evaluation episode the policy should be fixed to estimate (6). Can the author further explain how the network is updated?

Using evaluation to learn parameters seems unrealistic, the evaluation may happen only one time in practice. By evaluation, my understanding is that how much we can gain if we deploy such a policy. If the updating NERS requires to use evaluation data, this largely limited usability.

I am also confused by the statement “The replay reward is interpreted as measuring how much actions of the sampling policy help the learning of the agent for each episode.” Eq (6) says that the reward is actually how much it improved from the current evaluation to the previous one. No matter you use the special sampling method or not, it should make an improvement. So this difference does not indicate “how much the sampling distribution can help.”

In the empirical study, NERS does not show a clear benefit from the learning curves. I believe it is better to average over a smoothing window before averaging over random seeds. Or do more runs. Doing more runs should not be that computationally expensive at least on Pendulum and LunarLander. And one question, how do you implement the Prioritized ER? Do you also use the importance ratio to anneal the bias as described in section 3.4 in that paper (https://arxiv.org/pdf/1511.05952.pdf)? Do you ever tune a bit the parameter beta in that formulae?

An additional note about related work.
The sampling distribution is an important problem in RL and is not well/rigorously studied. I really think it worths a more complete discussion of related work. For example, the author might also discuss the Langevin dynamics Monte Carlo sampling method in RL (Frequency-based search control in Dyna by Pan et al.), as their sampling distribution is supported by intuition and suggestive theoretical evidence and they show their method is better than prioritized ER and ER.

---

> ### Author Response · Authors · 2020-11-15
> **Response to Reviewer #2 (2/2)**
>
> ---
> **Q4. I am also confused by the statement “The replay reward is interpreted as measuring how much actions of the sampling policy help the learning of the agent for each episode.” Eq (6) says that the reward is actually how much it improved from the current evaluation to the previous one. No matter you use the special sampling method or not, it should make an improvement. So this difference does not indicate “how much the sampling distribution can help.”**
>
> - We agree that the increase of the reward can be made both by the agent learning a better policy, or the experience sampler sampling more effective samples. Thus the reward may increase regardless of the sampling policy we use due to the training of the agent's policy, but note that we need to sample better transitions in order to obtain **larger increase**of the reward.
>
> ---
>
> **Q5. And one question, how do you implement the Prioritized ER? Do you also use the importance ratio to anneal the bias as described in section 3.4 in that paper (https://arxiv.org/pdf/1511.05952.pdf)? Do you ever tune a bit the parameter beta in that formulae?**
>
> - Yes we **do use $\alpha=0.5$ to anneal the bias**, which is smaller than $\alpha=0.7$ used in the original paper, and thus our PER sampler is less biased. We used this ratio since it is known to work well (Rainbow, Hessel et al. 2018).  We linearly increase $\beta$ from $0.4$ to $1$. Page 17 (Supplementary file) of the revision provides the detailed configurations of the PER.
>
> ---
> **Q6. An additional note about related work. The sampling distribution is an important problem in RL and is not well/rigorously studied. I really think it worths a more complete discussion of related work. For example, the author might also discuss the Langevin dynamics Monte Carlo sampling method in RL (Frequency-based search control in Dyna by Pan et al.), as their sampling distribution is supported by intuition and suggestive theoretical evidence and they show their method is better than prioritized ER and ER.**
>
> - We appreciate the insightful comment. We also agree that the sampling distribution is critical for model-based RL. We have stated it from the line 17 of the Related Work section in Page 10 (How to sample is also a
> crucial issue to model-based RL algorithms ...) in our revised manuscript.
>
> ---
>
> We sincerely thank you for your insightful comments since they helped us to further improve the discussions in the paper. Please let us know if there is anything else we should address, or misunderstood.
>
> ---
>
> [1] Schaul, Tom, et al. "Prioritized experience replay." arXiv preprint arXiv:1511.05952 (2015).
>
> [2] Zha, Daochen, et al. E"xperience replay optimization". In Proceedings of the 28th International Joint Conference on Artificial Intelligence. AAAI Press, 2019. p. 4243-4249.
>
> [3] van Hasselt, Hado P., Matteo Hessel, and John Aslanides. "When to use parametric models in reinforcement learning?." Advances in Neural Information Processing Systems. 2019.

---

> > ### Comment · AnonReviewer2 · 2020-11-24
> > **Thanks for your response.**
> >
> > Thanks for your response, and I updated my rating.

---

> ### Author Response · Authors · 2020-11-15
> **Response to Reviewer #2 (1/2)**
>
> We thank you for your time and effort in reviewing our paper, as well as constructive comments. We have revised the manuscript by faithfully reflecting your comments. We respond to your comments below:
>
> ---
> **Q1. In the original prioritized ER paper, they use a ratio to mitigate the biased sampling issue, did the authors ever visualize what the result would be (say in Figure 1) if you use that ratio?**
>
> - In the PER [1] paper, the authors used $\alpha=0.7$, which results in more biased sampling than our version of the PER, which used $\alpha=0.5$. Since PER performs more sampling as $\alpha$ increases, using the $\alpha$ in the original paper will result in even more severe biased sampling based on TD-errors.
>
> ---
> **Q2. Eq (2) the input feature can be highly non-stationary/unstable. For example, some of the variables may decrease all the time, and some others may increase all the time. Intuitively, training with such data should be very challenging. It looks like that in order to resolve the issue of biased sampling, the authors introduce an even more difficult task. Do the authors have some comments about this?**
>
> - We agree that predicting **precise values**from the inputs will be a highly difficult problem, as such is known to be very challenging in cases such as multi-agent RL. However, please note that our NERS **does not predict precise importance**but rather estimates the **relative importance**of samples chosen from the buffer. Since this is all the information we need. As shown in Figure 3(d) and Figure 5(a), such consideration of relative importance by NERS actually results in **more stable training**compared to ERO.
>
> ---
> **Q3. I am confused about how the sampling network is updated. In Algorithm 1, my understanding is that if the current time step is the end of an episode, then update the sampling network. Is it correct? Note that Algorithm 1 indicates that the actor, critic are updated at each time step. But the paper also says that the NERS is updated at each evaluation step and this means that throughout the evaluation episode the policy should be fixed to estimate (6). Can the author further explain how the network is updated? Using evaluation to learn parameters seems unrealistic, the evaluation may happen only one time in practice. By evaluation, my understanding is that how much we can gain if we deploy such a policy. If the updating NERS requires to use of evaluation data, this largely limited usability.**
>
> - Thank you for the insightful comment. We set the replay reward calculated by each evaluation since performing evaluation multiple times is not very difficult in standard environments.
>
> - However, as you mentioned, it will be difficult to compute the reward in environments where evaluation is allowed only once. To resolve this issue, we have slightly modified the replay reward such that NERS can use the reward obtained from cumulative rewards at each **training episode**. We report its performance in the experiments in Figure 4(b) and Figure 4(c) of the revision (NERS*). We can see that its performance is almost similar to original NERS on BipedalWalker-v3 and LunarLanderContinuous-v2. We thank you for the insightful suggestion as this will further enhance the usability of our method.

---

### Official Review · AnonReviewer1 · 2020-11-02
**A sound idea and evaluation**

**Rating:** 7
**Confidence:** 4

**Review:**

EDIT: The statements about ERO clarify the contribution considerably. 6-->7

The authors propose an adaptively sampling mechanism optimized for policy improvement (NERS). By incorporating minibatch-wide information into the sampling score (while maintaining permutation invariance), they are able to out-perform reasonable baselines on a wide range of tasks.

While NERS rarely beats other methods decisively, it has a strong showing across continuous and discrete action tasks and with a variety of off-policy learners. However, a few things would strengthen the empirical results. Some notation of spread should be reported on all of the Tables (e.g. standard error). Number of random seeds should also be mentioned. The reasoning behind the task selection should also be made explicit. The Atari subset used here is a bit unusual, particularly the choice to not use frame-skip. Investigating the sampling decisions of NERS is attempted in Figure 4, but further work should be done to provide evidence to the 'diversity of samples' claim. Ideally, NERS wouldn't just trade off TD error and Q-value over time, but also within each batch. Reporting something like the average minibatch Q-value/ TD-error standard deviation on NERS vs other methods would be nice. A qualitative evaluation akin to Figure 1 would also help guide intuition.

The off-policy RL related works section is a bit over-long, discussing things like the dueling architecture which don't seem to be overtly related apart from coming from the same sub-field. On the other side of things, having skimmed the ERO paper it is definitely the most-closely related, and as such deserves a bit more time spent on discussing the differences. For example, it is a bit unclear if the two sampling reward functions are different. An uncharitable reading of this paper would be that it is just an architecture tweak on top of ERO, and while the empirical results help dispel this idea, I think a more explicit comparison would still be useful.

A related point is that the reward function for the sampler is quite unclear (Equation 6). How are these expectation evaluated in practice? I'd assume it'd just be the difference of value functions before and after the update, but the appendix suggest a more involved computation that doesn't appear to have been made explicit anywhere.

Final small point, towards the end a bi-GRU is mentioned as being used and I can't see where that'd come into play. Perhaps just a typo?

Overall, I like this paper. Evaluating an idea across a variety of learning algorithms, observation and action spaces is no small feat, and the results are very solid. With a few tweaks and explanations this would be a very strong paper.

---

> ### Author Response · Authors · 2020-11-15
> **Response to Reviewer #1 (2/2)**
>
> ---
> **Q4. The off-policy RL related works section is a bit over-long. Having skimmed the ERO paper it is definitely the most-closely related, and as such deserves a bit more time spent on discussing the differences.**
>
> - We thank the reviewer for helpful suggestions. We have reduced the length of statements for off-policy RL in the Related Work section. We have also provided detailed discussion of the difference between NERS and ERO in the paragraph below Equation (7). We also summarize the differences between ERO and NERS below:
>
> - First of all, ERO learns the sampling rate for each individual transition with a simple MLP, without consideration of its relative importance over other samples. However this approach will score two near-redundant transitions to have similar importance. On the other hand, NERS learns the **relative importance**among all transitions in the given batch with a **permutation-equivariant set function** (Please see Figure 2).
>
> - Secondly, ERO performs two-stage sampling (Bernoulli sampling followed by random sampling), which is both ineffective as it requires $O(N)$ operations and inefficient since it results in sampling an **excessive number of redundant transitions**. Thus they use a random sampling to further reduce the samples but this makes ERO to behave similarly to that of the random sampling (Please see the paragraph above Eq.(5), in page 4). Contrarily, NERS performs prioritized sampling (from a Dirichlet distribution from the softmax output), which can be efficiently done in $O(logN)$ operations (using sum-tree). Further NERS performs importance sampling and places weights on the samples even after sampling (Eq.5).
>
> ---
> **Q5. How are these expectation evaluated in practice? I'd assume it'd just be the difference of value functions before and after the update, but the appendix suggest a more involved computation that doesn't appear to have been made explicit anywhere.**
>
> - We calculated the replay reward (Eq. (6)) as the difference in the **cumulative rewards**between the current and previous evaluations. The line 4-8 on page 18 in the supplementary material, we mentioned that each off-policy algorithm conducts evaluations at a fixed frequency, and we compute the replay reward at each evaluation step.
>
> ---
> **Q6. Final small point, towards the end a bi-GRU is mentioned as being used and I can't see where that'd come into play. Perhaps just a typo?**
>
> - This is indeed a typo and we have revised in in the updated manuscript. We apologize for the mistake, and thank you for the correction.
>
>
> ---
> [1] van Hasselt, Hado P., Matteo Hessel, and John Aslanides. "When to use parametric models in reinforcement learning?." Advances in Neural Information Processing Systems. 2019.

---

> ### Author Response · Authors · 2020-11-22
> **Response to Reviewer #1 (1/2)**
>
> We sincerely appreciate your time and efforts in reviewing our paper, as well as the constructive comments. We respond to each of your comments one by one.
>
> **Q1. The reasoning behind the task selection should also be made explicit. The Atari subset used here is a bit unusual, particularly the choice to not use frame-skip.**
>
> - Please note that we **do use frame-skip**of 4. For Atari tasks, we have used completely **the same configurations as the ones used in [1]**. For detailed configuration, please see training details in **Page 17**of our revised manuscript. We also have explained how we have chosen environments in the first subsection of Section 3.1 on page 5.
>
> - We have included additional results on **more Atari tasks** in Table 2 (Page 7) of our revised manuscript.
>
> ---
> **Q2. The number of random seeds should also be mentioned.**
>
> - We used $5$ random seeds. Although we already mentioned that five instances had been used, we will further clarify that this denotes five random seeds. Please see captions in Figure 3, 4, 5 (page 6-8) in our revised manuscript, respectively.
>
> ---
> **Q3. Investigating the sampling decisions of NERS is attempted in Figure 4, but further work should be done to provide evidence to the 'diversity of samples' claim. Ideally, NERS wouldn't just trade off TD error and Q-value over time, but also within each batch. Reporting something like the average minibatch Q-value/ TD-error standard deviation on NERS vs other methods would be nice. A qualitative evaluation akin to Figure 1 would also help guide intuition.**
>
> - We appreciate the reviewer’s suggestion. We agree that what you suggested will be an effective way to show the diversity, so obtained the following table (please see also Table 3 and the last paragraph of Section 3 in page 9 in our revised manuscript).  The table shows sampled transitions' statistical values for $Q$-values and TD-errors on Pendulum-v0 under SAC at 10,000 training steps with initially 1,000 random actions. It is easily observable that **NERS has  a higher standard deviation of TD-errors and $Q$-values than RANDOM and ERO.** Furthermore, NERS still has a higher average of TD-errors and $Q$-values than others. Although PER has the highest standard deviation for TD-errors, it also has the lowest standard deviation for $Q$-values. It means that PER sampled biased transitions.
>
> ---
> | Method 	| STDEV of TD-errors 	| STDEV of $Q$-values 	| AVG of TD-errors 	| AVG of $Q$-values 	|
> |:------:	|:------------------:	|:-----------------:	|:----------------:	|:---------------:	|
> |  NERS  	|       723.01       	|       65.22       	|       87.54      	|     -104.13     	|
> | RANDOM 	|       528.76       	|       60.46       	|       62.43      	|     -120.13     	|
> |   PER  	|       1256.71      	|       49.78       	|      139.49      	|     -138.05     	|
> |   ERO  	|       560.56       	|       59.16       	|       65.44      	|     -119.03     	|
>
> ---
> - Please note that if the diversity is low, it is intuitively difficult to maintain high values of TD-errors since the updated actor and critic networks **will decrease the values**, and the Q-value **will not increase** without observing new transitions.
>
> - Thus, NERS having both high Q-values and TD-errors in Figure 5 (Page 8 in the revision) over RANDOM suggests that NERS samples more diverse transitions which are effective to both the actor and the critic networks while training.
>
> - The **diversity of the samples** claim refers to NERS's **ability to** sample transitions with different criteria (focusing on either the Q-value, TD-errors, rewards, or even the raw states).
>
>
>
> ---
> [1] van Hasselt, Hado P., Matteo Hessel, and John Aslanides. "When to use parametric models in reinforcement learning?." Advances in Neural Information Processing Systems. 2019.

---

> > ### Comment · AnonReviewer1 · 2020-11-24
> > **Re: Frameskip**
> >
> > Thanks for clarifying about the usage of frameskip. I'm still a bit confused about the level descriptions, which include the phrase "NoFrameskip-v4". I'm guessing this means no frameskip is included in the environment by default, but then the 4 frameskip used is added afterwards? Not a big deal, but a bit confusing.

---

> > > ### Author Response · Authors · 2020-11-24
> > > **Answer for Frameskip**
> > >
> > >  We have now realized why the reviewer has been confused. The term **NoFrameskip-v4** appears in the supplementary material. For NERS under Rainbow, our implemented code first loads an environment with the NoFrameskip-v4 version  (please see line 15-16 of env.py in the uploaded code). After that, the environment does four times repeated actions whenever the step function is called (line 58-66 of env.py). Since it makes readers confused, we have revised it by removing the term in the supplementary material. Thank the reviewer for cooperating our manuscript more readable.

---

### Decision · Program_Chairs · 2021-01-07
**Final Decision**

**Decision:**

Accept (Poster)

**Comment:**

All reviewers agree that this paper is worth publishing. It investigates a novel idea on how to adaptively prioritise experiences from replay based on relative (within-batch) importance. The empirical investigation is thorough, and while the performance improvements are not stunning, the benefit is surprisingly consistent across many environments.